# Dynamic subcellular localization of a respiratory complex controls bacterial respiration

**François Alberge, Leon Espinosa, Farida Seduk, Léa Sylvi[†], René Toci, Anne Walburger, Axel Magalon***

Laboratoire de Chimie Bactérienne UMR7283, Institut de Microbiologie de la Méditerranée, Centre national de la recherche scientifique, Aix Marseille Université, Marseille, France

**Abstract** Respiration, an essential process for most organisms, has to optimally respond to changes in the metabolic demand or the environmental conditions. The branched character of their respiratory chains allows bacteria to do so by providing a great metabolic and regulatory flexibility. Here, we show that the native localization of the nitrate reductase, a major respiratory complex under anaerobiosis in *Escherichia coli*, is submitted to tight spatiotemporal regulation in response to metabolic conditions via a mechanism using the transmembrane proton gradient as a cue for polar localization. These dynamics are critical for controlling the activity of nitrate reductase, as the formation of polar assemblies potentiates the electron flux through the complex. Thus, dynamic subcellular localization emerges as a critical factor in the control of respiration in bacteria.

***For correspondence:** magalon@imm.cnrs.fr

**Present address:** [†]MIO, Aix Marseille Université, Marseille, France

**Competing interests:** The authors declare that no competing interests exist.

**Reviewing editor**: Gisela Storz, National Institute of Child Health and Human Development, United States

## Introduction

Respiration is an essential process for most living organisms. The free energy derived from oxidation of reducing equivalents generated by cell metabolism or taken up from the environment is converted into a protonmotive force (*pmf*) across a membrane that is used, among many processes, to drive adenosine triphosphate synthesis. Although oxygen is the most common terminal electron acceptor of respiratory chains, prokaryotes can make use of alternative acceptors in anoxic environments, such as nitrogen oxides, elemental sulfur and sulfur oxyanions, organic N-oxides, etc. In eukaryotes, respiration is confined to a specific organelle, the mitochondrion and more specifically to its inner membrane which displays remarkably intricate ultra-structural features that impact the respiratory output (*Cogliati et al., 2013*). Respiration involves multimeric membrane-embedded oxidative phosphorylation (OXPHOS) complexes which bear several metal cofactors through which electrons are transported. The organization of OXPHOS complexes in the inner mitochondrial membrane has been the subject of intense debate during the last decade after the discovery of supramolecular assemblies by the landmark study of Schägger in 2000 (*Schägger and Pfeiffer, 2000*), later substantiated by electron microscopy studies (for a review, see *Vonck and Schafer, 2009*). Plasticity of the almost linear mitochondrial OXPHOS chain is ensured by a dynamic equilibrium between isolated complexes and supramolecular assemblies which is, in turn, influenced by both specific lipids and stabilizing factors (*Zhang et al., 2002*; *Ikeda et al., 2013*). Such an equilibrium is likely beneficial for the function of the mitochondrion allowing the cell to respond to cellular and environmental cues. Moreover, the extraordinarily low degradation rate of OXPHOS complexes in mitochondria (*Price et al., 2010*; *Kim et al., 2012*; *Nelson et al., 2013*) reinforces the idea that the dynamic association into supercomplexes adds an effective switch to control electron flows in response to environmental changes (*Lapuente-Brun et al., 2013*). The flexibility of energy metabolism is also critical for the adaptation of a number of prokaryotes to varying environments

**eLife digest** Respiration occurs at different levels: the body, the organ, and the cells. At the cellular level, it is a molecular process that produces a high-energy molecule called adenosine triphosphate (ATP) using the biochemical energy stored in sugars, fatty acids, and other nutrients. Along with the ATP, this process also provides another source of energy to the cell: an electrochemical gradient across the membrane used for a range of processes ranging from the transport of molecules and ions to cell motility.

In order to thrive, cells need to quickly respond to cues from the environment or elsewhere in the cell. A cell must therefore have the ability to increase or decrease cellular respiration and the production of ATP to ensure it has an appropriate supply of energy. In bacteria, the protein complexes responsible for cellular respiration are embedded in the cell membrane. In the past decade, research has suggested that large molecules are arranged in a specific way throughout the bacterial cell, which directly influences how they work.

Alberge et al. tested this idea by studying the localization of a respiratory complex called nitrate reductase—which is important for generating energy in the absence of oxygen—through the introduction of a fluorescent marker tagged to the complex in the cell membrane of a rod-shaped bacterium called *Escherichia coli*. This allowed the complex to be tracked when the cells were viewed using a microscope. The experiments revealed that the location of the complex varies depending on how much energy the cell requires. For example, when the cells are in an oxygen-poor environment, the nitrate reductase complex moves towards the poles at each end of the bacterial cells. This allows the cells to produce ATP more efficiently through respiration of nitrate. Alberge et al. show that a 'proton gradient', caused by positively charged hydrogen ions moving through the cell membrane as the result of respiration, controls where the complexes are located in the membrane.

Alberge et al.'s findings provide experimental support that dynamic localization of respiratory complexes plays an important role in controlling respiration in bacteria. The next challenge will be to identify the genes that influence the distribution of respiratory complexes throughout the cell, which may help to explain how bacterial cells have adapted to specific environments.

(*Unden et al., 2014*). The utilization of modular OXPHOS chains confers flexibility depending on the available energy source or terminal electron acceptor but also allows modulation of the resulting *pmf*. So far, the adjustment of respiration to varying environments in prokaryotes is considered to be the result of an intricate transcriptional regulation network that controls the expression of OXPHOS complexes with varying electrogenic capacities.

At the same time, there is cumulative evidence for an elaborate spatial organization of macromolecules in bacterial cells. We thus hypothesized that the OXPHOS process could be regulated through the dynamic subcellular localization of complexes in response to the metabolic demand. Here, we investigate the native localization of a major OXPHOS complex under anoxic conditions, the quinol oxidizing nitrate reductase complex from the gut bacterium *Escherichia coli*, and its dynamics upon changes in the environment by real-time fluorescence imaging in live cells.

## Results

The *E. coli* nitrate reductase complex is composed of three subunits (NarGHI) and likely organized in a dimer under physiological conditions (*Bertero et al., 2003*). A di-heme *b*-type cytochrome subunit, NarI, ensures quinol oxidation and membrane anchoring of the cytoplasmically oriented catalytic dimer, NarGH, where nitrate reduction takes place. The C-terminus of the catalytic subunit NarG protruding from the quaternary structure of the complex was labeled with a green fluorescent protein (GFP). The hybrid gene was expressed under the control of its native promoter activated by FNR and NarL transcriptional factors in response to anaerobiosis and nitrate, respectively and ectopically integrated into the chromosome of the nitrate reductase-deficient strain JCB4023 (*Potter et al., 1999*). The corresponding fusion was functional as verified by cell growth in nitrate-respiring conditions where a functional nitrate reductase complex is mandatory (*Figure 1A*). Fractionation studies confirmed the membrane localization of the complex (*Figure 1B*). The in vitro assay of the functional integrity of the NarG-gfpH catalytic module showed that the activity was only

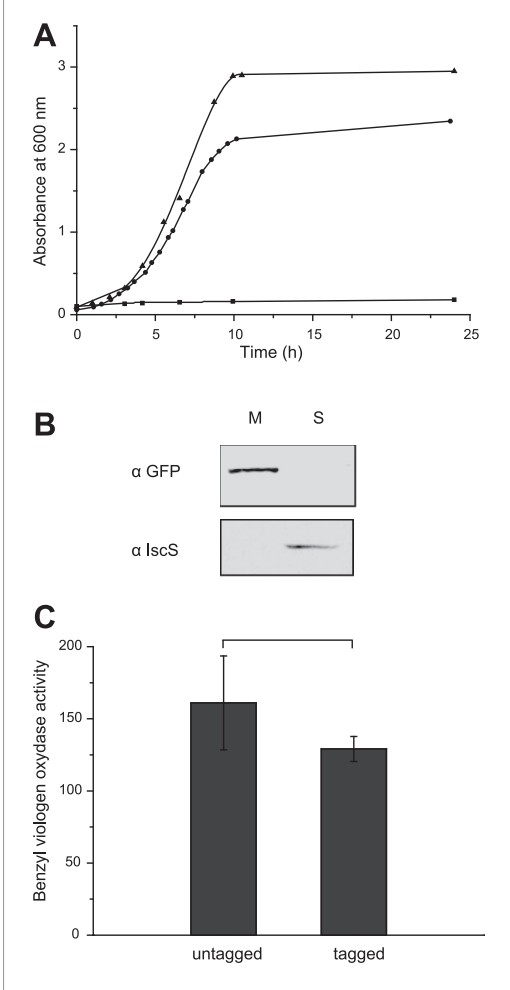

**Figure 1**. The GFP-labeled nitrate reductase complex is active and fully assembled .(**A**) Growth curves of *E. coli* strains expressing untagged or tagged-NarGHI under nitrate-respiring conditions. Cells were grown anaerobically in a minimal medium using glycerol as sole carbon source and nitrate as terminal electron acceptor. JCB4011 strain expresses the untagged NarGHI complex (▲), whereas the LCB3635 strain expresses the GFP-tagged NarGHI complex (●). As a negative control, the nitrate reductase-deficient strain JCB4023 shows no growth under these conditions (■). The estimated generation time is about 80 ± 10 min for JCB4011 and 110 ± 10 min for LCB3635. (**B**) The GFP-tagged NarGHI complex is correctly localized to the membrane. Western blots were performed with antibodies raised against eGFP or IscS as a marker of soluble proteins on soluble (S) or membrane (M) fractions prepared from nitrate-respiring cells. (**C**) The activity of the NarGH catalytic module is unaffected by the eGFP fusion. Benzyl viologen:nitrate oxidoreductase activity assays were performed on membranes prepared from JCB4011 (untagged version) or LCB3635 (GFP-tagged version) cells grown under nitrate-respiring conditions and expressed in µmoles of nitrite produced min$^{-1}$ mg$^{-1}$ of nitrate reductase.

slightly affected by the fusion (*Figure 1C*). Thus, the GFP-labeled nitrate reductase complex is active in the cytoplasmic membrane.

The subcellular localization of the anaerobic GFP-labeled OXPHOS complex was characterized by fluorescence microscopy imaging on exponentially growing *E. coli* cells under nitrate-respiring conditions (*Figure 2A*). The fluorescence signal was only detected in the cytoplasmic membrane in agreement with fractionation studies. Interestingly, the fluorescence signal appeared as clusters that predominantly concentrated at the cell poles in nearly 80% of the cells (*Figure 2B–D*). Furthermore, fluorescence clusters concentrate at the cell poles in nitrate-respiring cells independently of the cell length and not at the division septum (*Figure 2—figure supplement 1*). This suggests that there is no relationship between the position of the clusters and the cell cycle.

We next evaluated the impact of varying electron routes on the cellular localization of the GFP-labeled nitrate reductase complex. First, fluorescence imaging was performed on exponentially growing cells under oxygen-respiring conditions. Surprisingly, the fluorescence signal was evenly distributed along the cytoplasmic membrane under those conditions (*Figure 2A*). As compared to nitrate-respiring cells, fewer clusters are present and evenly distributed in oxygen-respiring cells (*Figure 2B–D*). Second, anaerobic respiration on alternative substrates such as fumarate or trimethylamine *N*-oxide (TMAO) was tested. It induces a significant polar localization of the fluorescence signal albeit lower than under nitrate-respiring conditions (*Figure 3*). Fluorescence imaging was eventually analyzed on anoxic fermenting growing cells using glucose as sole carbon source. Strikingly, the fluorescence was evenly distributed as observed under oxygen-respiring conditions (*Figure 3*). Detailed analysis of the fluorescence signal distribution in all these metabolic conditions revealed that the formation of fluorescent clusters is systematically associated with polar localization (*Figure 3—figure supplement 1*). Thus, the nitrate reductase complex displays a dynamic subcellular localization in response to the metabolic demand.

A major difference between the metabolic conditions studied above is the resulting *pmf*, based on the distinct proton transfer capabilities of the OXPHOS complexes involved. The present finding imparts particular significance to the previous reports that *pmf* can play a role in

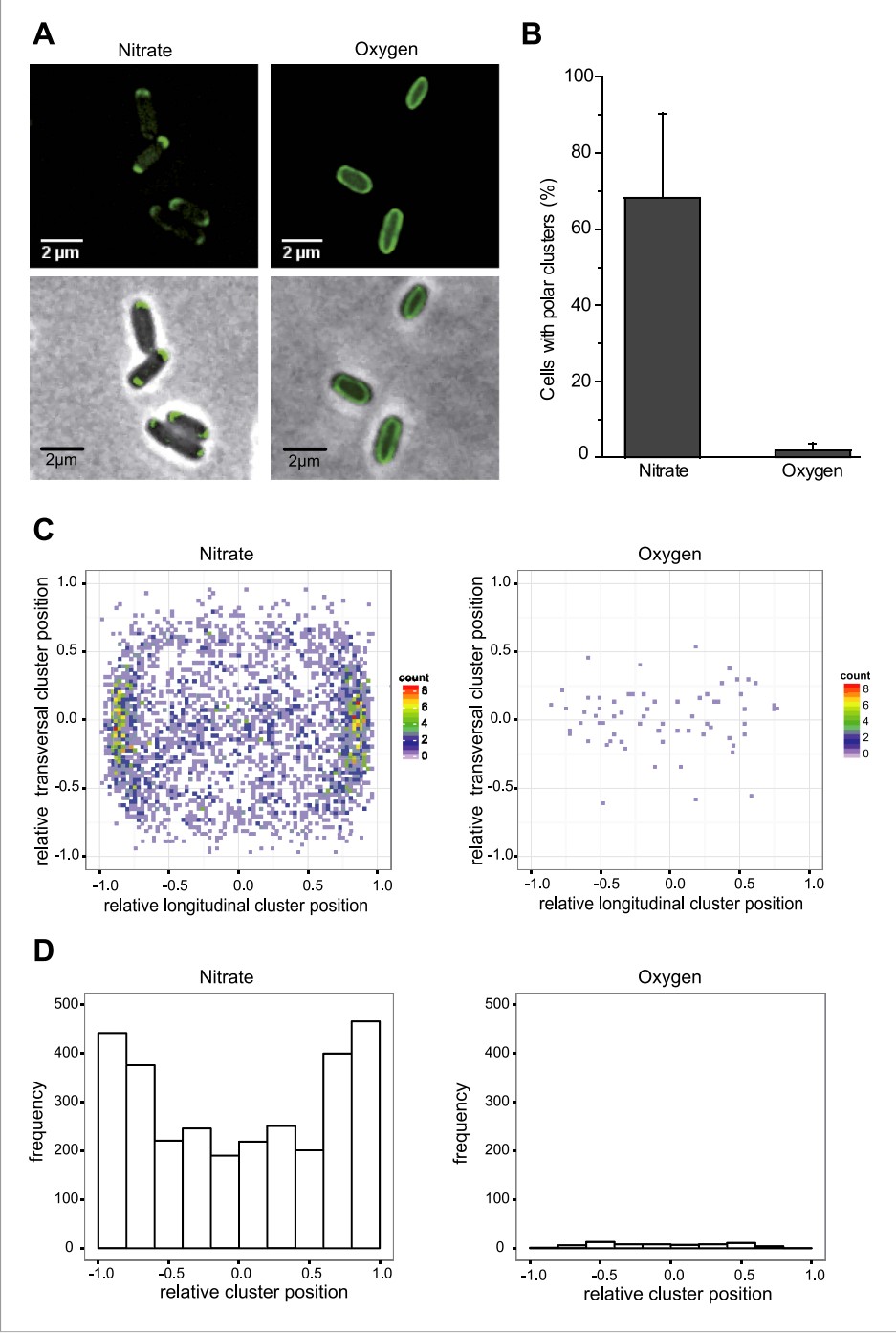

**Figure 2**. The GFP-labeled nitrate reductase complex concentrates at the cell poles under nitrate-respiring conditions. (**A**) Fluorescence images (top) and overlays of fluorescence and phase contrast images (bottom) are shown for nitrate-respiring and oxygen-respiring cells. The deconvolved image of the fluorescence signal is shown in green, and the cell outline is shown by phase contrast. (**B**) Mean frequency of cells displaying clusters at the cell poles for nitrate-respiring or oxygen-respiring cells. (**C**) A density map was built from a two-dimensional histogram (relative transversal cluster position vs relative longitudinal cluster position) of the fluorescence signal observed in nitrate-respiring or oxygen-respiring LCB3635 cells. The color map shows the interpolated density of clusters positions. The dots represent the individual clusters (a little jittering was added to avoid overlapping of the dots). (**D**) The histogram of the fluorescence signal clusters across the transversal axis of nitrate-respiring or oxygen-respiring LCB3635 cells is shown. In both conditions, more than 500 cells were analyzed. Under

*Figure 2. continued on next page*

*Figure 2. Continued*

nitrate-respiring conditions, a strong enrichment of the clusters is observed at the cell poles. Under oxygen-respiring conditions, only few cells exhibit clusters which have an even distribution along the cell axis.

The following figure supplement is available for figure 2:

**Figure supplement 1**. Spatial distribution of clusters.

---

protein localization in bacteria (*Alcock et al., 2013*; *Rose et al., 2013*). Saliently, the dissipation of the electric component of the *pmf*, the transmembrane potential ($\Delta\Psi$), hampers proper localization of proteins involved in cell division, chromosome segregation or cell shape regulation (*Strahl and Hamoen, 2010*). However there is, to our knowledge, no report of a protein localization that would be specifically controlled by the other component of the *pmf*, the proton concentration gradient ($\Delta$pH). To evaluate the participation of the *pmf* in the subcellular localization of the GFP-labeled nitrate reductase and analyze the contribution of its two components, we first treated nitrate-respiring cells with distinct ionophores. As shown in *Figure 4A*, the addition of carbonyl cyanide *m*-chlorophenyl hydrazone (CCCP), a specific proton-ionophore rapidly dissipating the *pmf*, resulted in a significant reduction of cells displaying polar clusters. Unexpectedly, depolarization of the membrane potential through the addition of the potassium-ionophore valinomycin had no effect, whereas dissipation of $\Delta$pH by the electroneutral anion/OH$^-$-exchanger trichlorocarbanilide (TCC) (*Ahmed and Booth, 1983*) gave rise to significant delocalization of the fluorescent signal. To substantiate the critical role of $\Delta$pH for polar localization, we reasoned that artificial establishment of a proton concentration gradient by the light-driven proton translocation activity of the proteorhodopsin (PR) (*Walter et al., 2007*) should be sufficient to promote the polar recruitment of the nitrate reductase complex under anoxic fermentative conditions. Indeed, under fermenting conditions, it is considered that the $\Delta$pH is very low as compared to respiring conditions. As shown in *Figure 4B*, heterologous expression of PR resulted in polar localization of the fluorescence signal in nearly 40% of cells. Thus, we conclude that the proton concentration gradient is a critical cue for polar localization of this anaerobic OXPHOS complex. We next questioned the importance of nitrate reductase activity on its subcellular localization. To this end, a catalytically inactive but stable variant was used. In particular, the H50S substitution in NarG precludes insertion of two metal centers at the active site while x-ray structural analysis of the NarG$_{H50S}$HI complex revealed the absence of structural changes at the protein surface (*Magalon et al., 1998*; *Rothery et al., 2010*). Under nitrate-respiring conditions, the fluorescence signal associated with the H50S variant is uniformly distributed likely due to its inability to generate a $\Delta$pH (*Figure 5A,B* and *Figure 5—figure supplement 1*). In contrast, adding fumarate gives rise to a significant polar localization of the inactive GFP-labeled nitrate reductase complex reinforcing the conclusion that activity of the complex is essential for polar positioning unless a $\Delta$pH is established (*Figure 5A,B* and *Figure 5—figure supplement 1*). Interestingly, polar localization requires both the absence of oxygen and the establishment of a proton concentration gradient to which the nitrate reductase complex may contribute fuelling the localization mechanism.

Bacteria exploit the branched character of their OXPHOS chains to respond to varying environments with a great metabolic and regulatory flexibility (*Unden et al 2014*). The above-described modification of the localization pattern of the nitrate reductase complex in response to metabolic changes begs the question of its timeframe. First, nitrate-respiring cells were submitted to strong aeration and the fluorescent pattern was followed during the course of the anoxic–oxic transition (*Figure 6A*). While 80% of the bacterial cells initially displayed a pronounced polar localization of the GFP-labeled nitrate reductase complex, the signal started to change noticeably after 15 min, reaching an almost uniform distribution after one hour of aerobic transition. A similar temporal behavior has been observed under the reverse conditions using aerobically growing cells transferred to anoxic conditions (*Figure 6A*). In this case, more than 70% of the bacterial cells displayed a polar localization of the fluorescent signal after the transition. Thus, redistribution of the OXPHOS complex triggered by metabolic changes operates on timescales of several tens of minutes. While such timing may reflect the slow diffusion of such large membrane-embedded complexes in the crowded cytoplasmic membrane, we next questioned whether this switch is faster than the changes in

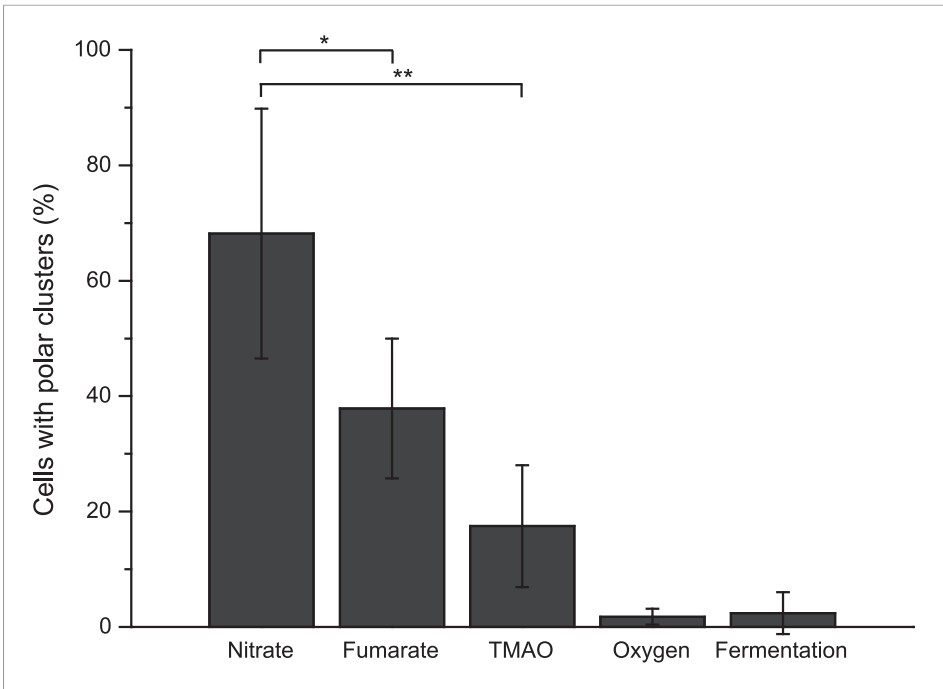

**Figure 3**. Metabolism-dependent localization of the nitrate reductase OXPHOS complex. Mean frequency of LCB3635 cells displaying clusters at the cell poles for various metabolic conditions (anaerobic respiration with either nitrate, fumarate or TMAO as terminal electron acceptor, aerobic respiration and anoxic fermenting conditions). Non-fermentable glycerol was used as sole carbon and electron source in all cases with the exception of fermenting conditions where it was replaced for glucose. Polar localization is only observed under anaerobic respiration whatever the terminal electron acceptor used.

The following figure supplement is available for figure 3:

**Figure supplement 1**. Comparative analysis of cells displaying clusters at the cell poles vs cells displaying clusters along the cell body upon varying metabolic conditions.

nitrate reductase protein content as the result of transcriptional regulation. At first, western-blotting analyses showed nearly unchanged level of GFP-labeled nitrate reductase complex upon anoxic–oxic transition despite the rapid redistribution of the complex in the cytoplasmic membrane (*Figure 6—figure supplement 1A*). In comparison, an increased level of complex is observed during the oxic–anoxic transition as the result of FNR-mediated regulation. To substantiate these observations, localization upshift experiments were reproduced in cells pre-treated with chloramphenicol (Cm), a protein synthesis inhibitor (*Figure 6B* and *Figure 6—figure supplement 1B*). During the anoxic–oxic transition, Cm had nearly no impact on the delocalization kinetics with only slight diminution of the nitrate reductase content. Interestingly, during the oxic–anoxic transition, the fluorescence signal remains uniformly distributed with a concomitant decrease of nitrate reductase content. To rule out any impact on the yielded *pmf* value under those conditions, the oxic–anoxic upshift experiment was reproduced in presence of PR. As shown in *Figure 6B*, establishment of a ΔpH was not sufficient to promote polar localization. Altogether, these results support the idea that induction of the expression of at least one gene during the oxic–anoxic transition promotes polar localization of the nitrate reductase complex.

The GFP-labeled nitrate reductase complex displays a dynamic localization pattern leading to the formation of discrete domains in the cytoplasmic membrane. At first, we reasoned that distinct subcellular localization may influence the intrinsic activity of the OXPHOS complex. The activity of the GFP-labeled complex was therefore assessed in membrane vesicles issued from cells that displayed either an even distribution (oxygen-respiring condition) or a strong polar enrichment (nitrate-respiring condition) of the complex. As shown in *Figure 7A*, we found no significant variation in the specific

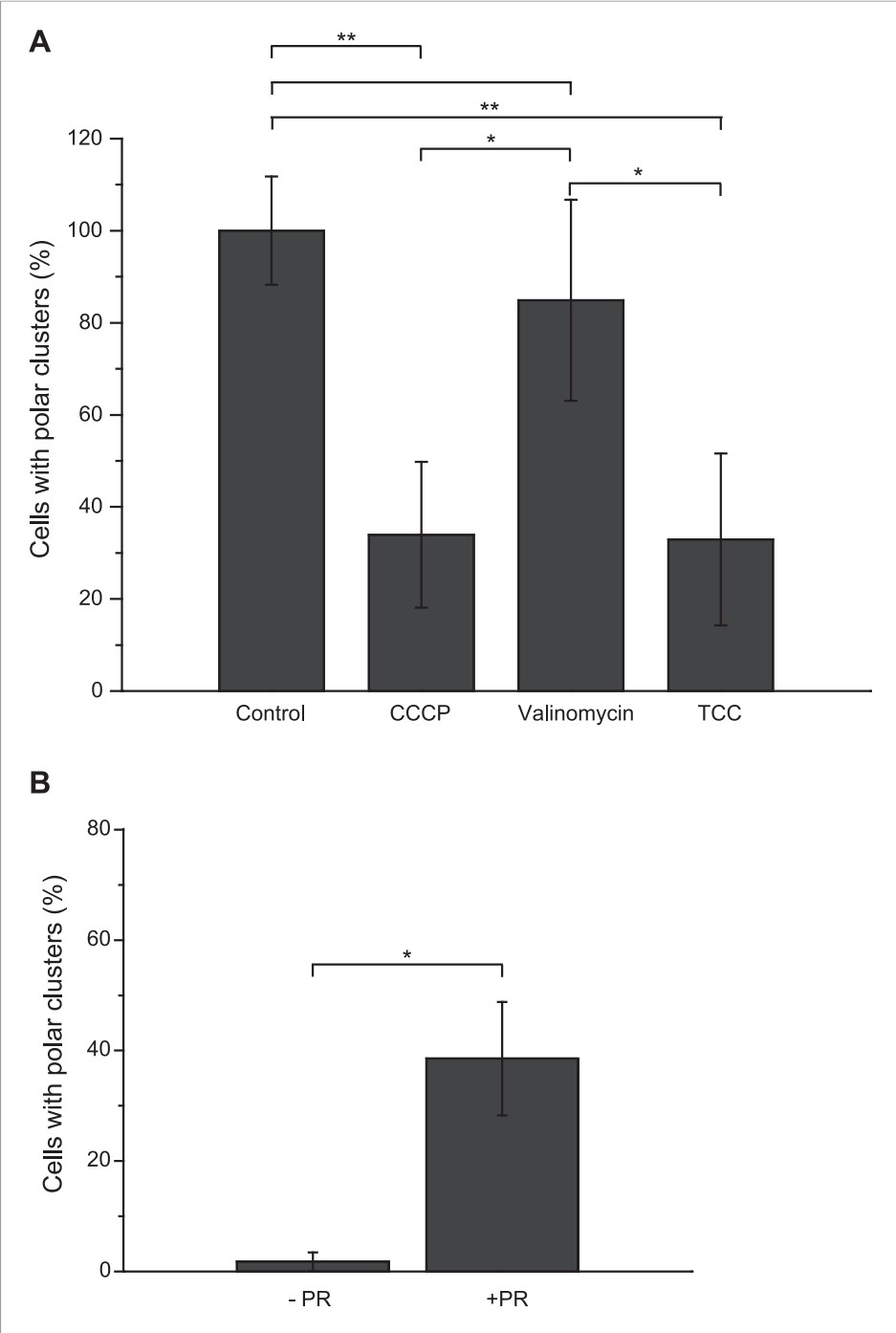

**Figure 4**. Proton gradient is a cue for polar localization of the anaerobic OXPHOS complex. (**A**) The *pmf* is important for polar localization of the nitrate reductase complex. Shown is the mean frequency of nitrate-respiring LCB3635 cells displaying clusters at the cell poles upon addition of the indicated ionophores. Each value was normalized with respect to that of the untreated cells (labeled as control) and expressed in % of the control. Images were taken 15 min after the treatment with the ionophore. (**B**) Establishment of an artificial proton concentration gradient in anoxic fermenting growing cells is sufficient to restore the polar localization. Shown is the mean frequency of fermenting growing LCB3635 cells displaying clusters at the cell poles in the absence (-PR) or presence of the proteorhodopsin (+PR).

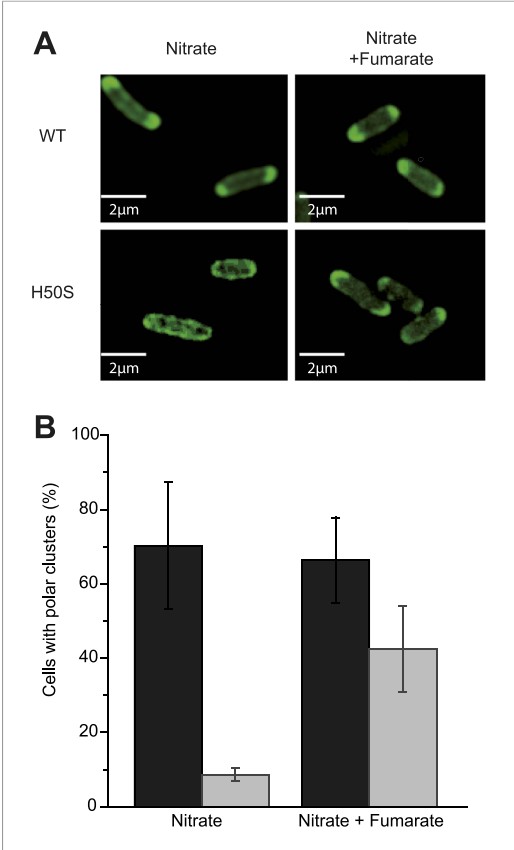

**Figure 5**. Activity of the nitrate reductase OXPHOS complex is not essential for polar positioning unless a ΔpH is established. (**A**) Fluorescence images of JCB4023/pVA70GFP cells expressing active (top) or inactive NarG-H50S variant of the GFP-labeled nitrate reductase (bottom) are shown. Cells were grown anaerobically either in glycerol-nitrate (nitrate) or glycerol-nitrate-fumarate (nitrate + fumarate) medium. (**B**) Mean frequency of cells displaying clusters at the cell poles under nitrate or nitrate–fumarate conditions. (Black bars) JCB4023/pVA70GFP cells expressing the active complex. (Gray bars) JCB4023/pVA70GFPH50S cells expressing the inactive variant.

The following figure supplements are available for figure 5:

**Figure supplement 1**. Activity of the nitrate reductase OXPHOS complex is not essential for polar positioning unless a ΔpH is established.

**Figure supplement 2**. Cells expressing the GFP-labeled nitrate reductase complex from the pVA70GFP plasmid show identical metabolism-dependent localization of the complex.

activities indicating that subcellular localization has no influence on the intrinsic activity of the OXPHOS complex.

We next hypothesized that the formation of discrete domains through the polar recruitment of the nitrate reductase may influence the overall yield of the electron transport chain. To evaluate the impact of polar localization on the electron flux from primary dehydrogenases to GFP-labeled nitrate reductase, it was essential to ensure an identical composition of the cytoplasmic membrane in terms of OXPHOS complexes. To account for this issue, we used fermenting-growing cells expressing or not PR which have an identical OXPHOS proteome but display distinct localization patterns of the GFP-labeled complex (*Figure 4B*). Upon addition of nitrate in the growth medium, electron flux through the nitrate reductase complex could be kinetically resolved by quantifying nitrite in the cell culture. As seen in *Figure 7B*, within the first 10 min following nitrate addition, the rate of nitrite production is significantly higher in PR-expressing cells than in control cells, indicating a direct correlation between subcellular localization and electron flux through the nitrate reductase complex. After one hour, the level of nitrite produced is nearly two orders of magnitude higher in PR-expressing cells. We conclude that environmental conditions (anaerobiosis and a ΔpH) promoting polar clustering of the nitrate reductase complex result in a higher efficiency of the associated respiratory chains.

## Discussion

In the past decade, the emerging field of bacterial cell biology has underscored the fact that dynamic subcellular localization is intimately linked to the biological function allowing control of fundamental processes, such as cell division, virulence, motility, or signal transduction (for reviews, see *Kiekebusch and Thanbichler, 2014*; *Laloux and Jacobs-Wagner, 2014*; *Nevo-Dinur et al., 2012*; *Shapiro et al., 2009*). Similarly, we have shown that the nitrate reductase respiratory complex is submitted to a spatiotemporal regulation in response to environmental conditions which, in turns, potentiates the electron flux through the associated respiratory chain. Nitrate respiration in *E. coli* is thus controlled by specific subcellular localization of its terminal reductase. Segregation of respiratory complexes within the cytoplasmic membrane has also been reported for components of bacterial aerobic chains (*Johnson et al., 2004*; *Lenn et al., 2008*; *Werner et al., 2009*; *Rexroth et al., 2011*; *Llorente-Garcia et al., 2014*). The actual view is that functional consequences are associated with a high level of structural organization of respiratory

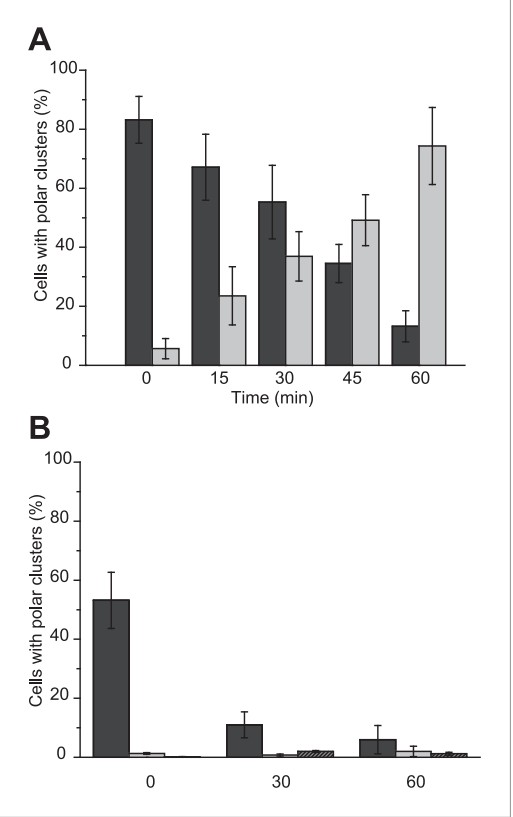

**Figure 6**. Metabolism-dependent localization changes occur in a timeframe of several tens of minutes. (**A**) (Black bars) Nitrate-respiring JCB4023/pVA70GFP cells were submitted to strong aeration and the fluorescence pattern was evaluated at 15-min intervals. (Gray bars) Oxygen-respiring JCB4023/pVA70GFP cells were shifted to anoxic conditions in presence of nitrate. In both cases, shown is the mean frequency of cells displaying clusters at the cell poles. (**B**) (Black bars) Upon chloramphenicol (Cm) addition, nitrate-respiring 3635/pBAD24 cells were submitted to strong aeration and the fluorescence pattern was evaluated at 30-min intervals. (Gray bars) Upon Cm addition, oxygen-respiring 3635/pBAD24 cells were shifted to anoxic conditions in presence of nitrate. (Shaded bars) Upon Cm addition, oxygen-respiring 3635/pPR cells were shifted to anoxic conditions in presence of nitrate. In all cases, shown is the mean frequency of cells displaying clusters at the cell poles.

The following figure supplement is available for figure 6:

**Figure supplement 1**. Metabolism-dependent localization changes occur in a timeframe of several tens of minutes.

complexes within bioenergetic membranes (for a review, see *Genova and Lenaz, 2014*). Interestingly enough, recent studies have evidenced the existence of functional micro-domains in bacterial membranes organizing a specific subset of proteins in space and time surmising a gain-of-function (*Lopez and Kolter, 2010*). In the same line of thought, it has recently been shown that clustering of multiple enzymes belonging to the same metabolic pathway into agglomerates accelerates the processing of intermediates (*Castellana et al., 2014*). Clearly, several alternatives can be found for the reported enhanced electron flux in the nitrate reductase-associated respiratory chains such as physical association into supercomplexes or segregating respiratory complexes into microdomains. In both cases, one would hypothesize that the probability of the quinone molecules to encounter the nitrate reductase is higher. Evidences have been provided for such quinone channeling within supercomplexes while the exact mechanism of channeling is not yet understood (for a review, see *Genova and Lenaz, 2014*).

By providing the first evidence for a spatio-temporal regulation of a respiratory complex in response to environmental conditions, this work establishes a basis for a deeper analysis on how environmental signals are translated into subcellular localization of respiratory complexes to adjust the respiration output.

## Materials and methods

### Bacterial strains and growth conditions

The *E. coli* strains and plasmids are described in Table1, *Supplementary file 1*. *E. coli* strains were grown aerobically at 37°C in defined minimal medium supplemented with 140 mM of glycerol used as sole carbon source and 100 mM nitrate. Anaerobic growth of bacteria is performed in gas tight hungate tubes under Ar atmosphere. For anaerobic growth under respiring conditions, nitrate, fumarate or TMAO were added at 100 mM final concentration and used as terminal electron acceptors. For anoxic fermentative growth, glycerol was replaced by glucose at 40 mM final concentration. The minimal medium is composed of potassium phosphate buffer (100 mM) adjusted to pH 7.4, ammonium sulfate (15 mM), NaCl (9 mM), magnesium sulfate (2 mM), sodium molybdate (5 μM), Mohr's salt (10 μM), and calcium chloride (100 μM). After filtration, casaminoacids (0.5%) and thiamine (0.01%) were added just before use together with antibiotics, if necessary. The nitrate reductase-deficient JCB4023 strain was used as recipient for integration of the translational *narG-egfp* fusion at the

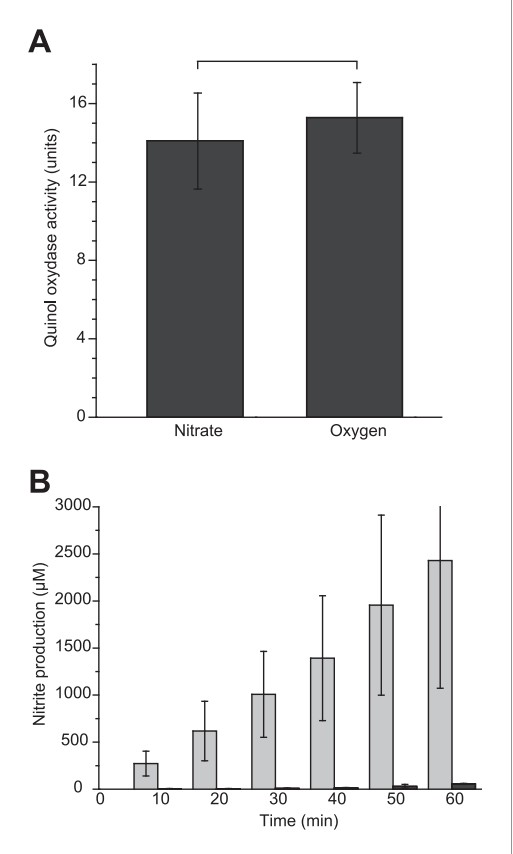

**Figure 7**. Polar localization determines integration of the nitrate reductase complex in anaerobic respiratory chains. (**A**) Constant activity of the GFP-tagged complex upon distinct subcellular localization. Quinol:nitrate oxidoreductase activity was measured on membranes prepared from oxygen-respiring or nitrate-respiring JCB4023 cells expressing the GFP-tagged NarGHI complex from the pVA70GFP plasmid. Activities are expressed in μmoles of nitrite produced min$^{-1}$ mg$^{-1}$ of nitrate reductase. (**B**) Enhanced nitrite production in cells with enforced polar localization of the GFP-NarGHI complex. At time 0, nitrate (100 mM final concentration) was added to anoxic fermenting growing LCB3635 cells expressing (+PR, light grey) or not expressing (−PR, black) the proteorhodopsin. Subsequently, nitrite production was detected in the cell culture over time using the Griess reaction. The indicated values were derived from raw data using a suitable standard curve and represented the means of three distinct experiments.

chromosomal $att_{\Phi 80}$ site using the procedure described in (*Haldimann and Wanner, 2001*). Introduction of the eGFP at the C-terminus of NarG was performed as described below. The oligonucleotides used in this study are described in Table 2, *Supplementary file 1*.

## Plasmid construction

To generate pVA70XN, the 3′ region of *narG* was PCR-amplified with primers 584 and 585, whereas the 5′ region of *narH* was amplified with primers 586 and 447. Both PCR products were then used as templates for another PCR reaction with primers 584 and 447 yielding a 1200 bp product including the 3′ end of *narG* followed by *Xma*I and *Not*I restriction sites and the native ribosome binding site of *narH*. The reaction product was then restricted with *Mun*I and *Aat*II, and the resulting fragment was ligated into the pVA70 vector which had been cut with the same enzymes. The entire cloned region was verified by sequencing.

To create pVA70GFP, which allows fusion of *egfp* to the 3′ region of the *narG* gene, the *gfp Xma*I-*Not*I fragment was excised from pEGFP-N1 (Clontech) and subsequently ligated into pVA70XN cut with the same enzymes. This plasmid allows the production of the GFP-tagged NarGHI complex.

To obtain pFA, which allows integration of a synthetic *narG-egfp narHJI* operon at the chromosomal $att_{\Phi 80}$ site, the entire operon including the p*nar* promoter was excised from pVA70GFP by restriction with *Sac*I and *Sal*I. The fragment was then ligated into the pAH162 CRIM plasmid which had been cut with the same enzymes. The pFA plasmid was then integrated at the $att_{\Phi 80}$ site of the nitrate reductase-deficient strain JCB4023 (*Potter et al., 1999*) according to the procedure described in *Haldimann and Wanner (2001)*. The resulting LCB3635 strain was maintained in the presence of 6 μg/ml of tetracycline.

## Cell fractionation

Cells were harvested in late-exponential phase, washed and resuspended in 40 mM Tris–HCl (pH 7.4), 1 mM MgCl$_2$. Bacterial cells were broken by passage through a French press. After an initial centrifugation at 20,000×*g*, differential ultracentrifugation at 250,000×*g* allowed the separation between soluble and membrane fractions which were frozen in liquid nitrogen and stored at −80°C until use.

## Enzyme activity and protein quantification

Nitrate reductase activity was measured with standard assays using reduced benzyl viologen or menadiol as electron donors (*Jones and Garland, 1977*; *Giordani et al., 1997*). As the result of chromosomal expression, a low level of the NarGHI complex is observed in membrane vesicles of the LCB3635 strain grown under oxygen-respiring condition. This situation precludes accurate assessment

of the quinol:nitrate oxidoreductase activity measured at 260 nm. Therefore, the JCB4023 strain was transformed with pVA70GFP, resulting in higher production of GFP-tagged NarGHI (eightfold as estimated by immunoblotting, data not shown) and allowing quinol activity measurements. The metabolism-dependent localization of the nitrate reductase was unaffected by the increased level of GFP-tagged NarGHI complex in the cells (*Figure 5—figure supplement 2*). The NarGHI protein concentration was estimated using rocket immunoelectrophoresis as described in *Lanciano et al. (2007)*. Western-blots were performed using antibodies raised against eGFP and IscS on samples run on 10% SDS-polyacrylamide gels. Quantitative analysis of the fold change in GFP-labeled nitrate reductase protein levels was achieved by integration of the eGFP chemiluminescence signal normalized with the IscS one using samples containing an equal cell mass.

Nitrite levels were determined spectrophotometrically by the Griess reaction. Briefly, 1 ml of the anaerobic cell culture was centrifuged. 500 µl of the cell-free supernatant was mixed with 100 µl of Griess reagent (0.5% of sulfanilic acid) and incubated for 5 min at room temperature. After addition of an equal volume of 0.6% of *N*-1-napthylethylenediamine dihydrochloride, the solution was incubated in the dark for 30 min before measuring the absorbance at 525 nm. A standard curve was made with a nitrite solution allowing quantitative measurement.

## Fluorescence microscopy

For fluorescence microscopy, cells were grown aerobically or anaerobically to midexponential phase at 37°C, and 2 µl was mounted on microscope slides covered by a thick fresh minimal medium agar pad. In case of anaerobically growing cells, images were taken after a 5–10 min delay shown to be optimal for activation of the GFP moiety during the mounting process. Furthermore using this procedure, we have noticed that the localization pattern was unchanged after 1 hr under the agar pad. The slide was analyzed by microscopy using a Nikon Eclipse TiE PFS inverted epifluorescence microscope (100 × oil objective NA 1.3 Phase Contrast) and a Hamamatsu OrcaR2 CCD camera. Images were collected with NIS elements software. Observation of the fluorescence signal of GFP-labeled nitrate reductase under aerobic conditions was performed on cells grown in minimal medium with glycerol as sole carbon source and with nitrate for NarL-mediated *nar* operon expression. Under this condition, oxygen is preferred to nitrate as terminal electron acceptor (*Unden et al, 2014*).

For evaluation of the participation of the *pmf* in the polar localization of the GFP-tagged NarGHI, ionophores were added to nitrate-respiring cells at midexponential phase and images were taken after 15 min of incubation. CCCP, valinomycin, and TCC were used at 100, 30, and 200 µM final concentrations, respectively according to (*Ahmed and Booth, 1983*). To build an artificial proton gradient across the cytoplasmic membrane, PR was expressed in LCB3635 cells using the pPR plasmid (*Tipping et al., 2013*). Cells transformed with the pBAD24 plasmid were used as control. Transformants were grown under aerobic conditions in a minimal medium supplemented with glucose and ampicillin. At early log phase, PR expression was induced with 0.02% arabinose together with *all-trans*-retinal (10 µM). At an OD600 of 1, the culture was shifted to anaerobic conditions and exposed to light for one hour, allowing the establishment of a proton gradient under these anoxic fermenting conditions. Fluorescence images were then taken as described above. Thanks to the red fluorescence emission of PR (*Beja et al., 2000*), uniform distribution of the fluorescence signal is observed at the cell membrane using appropriate filter thus confirming its membrane localization (data not shown). Upon establishment of a polar localization of the GFP-tagged NarGHI complex in PR-expressing cells, nitrate was added at a final concentration of 100 mM and the rate of nitrite production was then determined spectrophotometrically by the Griess reaction as described above.

## Image and statistical analysis

All the objects detection and quantification in images were performed with a new Fiji/ImageJ plugin developed specifically for the treatment of microscopic images of bacterial cells. This plugin is called MicrobeJ and was created by A Ducret in Y Brun laboratory (http://www.indiana.edu/~microbej/index.html) with Fiji software. Schematically, the clusters contrast was enhanced by a FFT band pass filter or by a morphological Top Hat filtering. The procedure includes automatic detection of cell shapes, medial axis determination, measurement of the cell size, and determination of the local fluorescence maxima (cluster detection) and of their absolute and relative position. The results of the image analysis were treated with R software (R Core Team [2014]. R: A language and

environment for statistical computing. R Foundation for Statistical Computing, Vienna, Austria. URL http://www.R-project.org). An R script yields the number of cells containing at least one cluster with a relative position < −0.6 or > +0.6. These regions were defined as the cell poles (*Figure 2—figure supplement 1B*). The density map was calculated with the R ggplot2 package (H. Wickham. ggplot2: elegant graphics for data analysis. Springer New York, 2009). The ratio of cells with a least one polar cluster over the total number of cells shown in all the figures was calculated as the mean of at least three independent experiments. In each case, more than 500 cells were analyzed from three independent experiments. In drug treatment experiments, the effect was compared against paired control experiments. All statistical tests were performed by the no parametric Wilcoxon/Mann–Whitney method when appropriate. Tests yielding a p value >0.05 were assumed as non-significant difference. One star stands for a p value ≤0.05 and two stars stand for a p value ≤0.01.

## Acknowledgements

We thank all the reviewers for their comments to improve the manuscript. We thank T Mignot, A Ducret, L Loiseau, B Ezraty, and T Doan (Laboratoire de Chimie Bactérienne, Marseille, France) for providing materials and strains. This work was supported by the Centre National de la Recherche Scientifique and the Agence Nationale de la Recherche. LS was supported by a fellowship from the Agence Nationale de la Recherche (ANR SPINFOLD n°09-BLAN-0100). The authors are also grateful to the Biophotonic facility available at the Laboratoire de Chimie Bactérienne and the Centre National de la Recherche Scientifique, the Fondation pour la Recherche Médicale for financial support in the acquisition of instrumentations.

## Additional information

### Funding

| Funder | Grant reference | Author |
| --- | --- | --- |
| Agence Nationale de la Recherche | 09-BLAN-0100 | Léa Sylvi |
| Ministère de l'Enseignement supérieur et de la Recherche | PhD Fellowship | François Alberge |
| Centre National de la Recherche Scientifique | | Leon Espinosa, Farida Seduk, René Toci, Anne Walburger, Axel Magalon |

The funders had no role in study design, data collection and interpretation, or the decision to submit the work for publication.

### Author contributions

FA, LE, AW, Conception and design, Acquisition of data, Analysis and interpretation of data, Drafting or revising the article; FS, LS, RT, Conception and design, Acquisition of data, Analysis and interpretation of data; AM, Conception and design, Analysis and interpretation of data, Drafting or revising the article

## Additional files

### Supplementary file

• Supplementary file 1. List of strains, plasmids, and oligonucleotides.

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
