## [Decision Letter]

Thank you for sending your work entitled “Dynamic subcellular localization of a respiratory complex controls bacterial respiration” for consideration at *eLife*. Your article has been favorably evaluated by Michael Marletta (Senior editor), a Reviewing editor, and three reviewers. The Reviewing editor and the reviewers discussed their comments before we reached this decision, and the Reviewing editor has assembled the following comments to help you prepare a revised submission.

The reviewers thought the data clearly showed nitrate reductase complex exhibits dynamic localization in response to changes in the transmembrane proton gradient. However, they felt the following changes are critical for the manuscript to be considered for *eLife*.

1) Additional experiments are needed to support the conclusion that “these dynamics are critical for controlling the activity of nitrate reductase.”

A mutant that does not localize properly and consequently is inactive should be identified.

Blue-Native Gel Electrophoresis should be carried out to test the inference that super-assemblies of respiratory complexes (such as observed for mitochondria) are formed. The co-association of other components should also be tested in these assays as well as by fluorescence and immune-labeling microscopy.

The claim that the changes in localization are quicker than would be expected if changes in gene expression were responsible [even though bacterial genes are induced within minutes of anaerobiosis (see Li et al. 1987 J. Bacteriol. 169:4614-4620)], should be tested by localization upshift experiments in cells pre-treated with a protein synthesis inhibitor. The authors should also examine the localization of proteorhodopsin and confirm the localization of native or functional epitope tagged NarG by immunolocalization in these experiments. Finally, there needs to be a negative control to show that proteorhodopsin does not direct any GFP-tagged membrane protein into bi-polar clusters.

2) Appropriate statistical analysis of data needs to be included for all experiments.

3) The Discussion needs to be expanded and address the question “How do the findings fit into the wider field of bacterial cell biology, metabolic channeling in bacteria and organelles?” (Werner et al. PNAS 2009) rather than speculate on host-pathogen interactions in the gut.

[Editors' note: further revisions were requested prior to acceptance, as described below.]

Thank you for resubmitting your work entitled “Dynamic subcellular localization of a respiratory complex controls bacterial respiration” for further consideration at *eLife*. Your revised article has been favorably evaluated by Michael Marletta (Senior editor), a Reviewing editor, and two reviewers. The manuscript has been improved but there are some remaining issues that need to be addressed before acceptance, as outlined below:

1) The new result with the inactive mutant that does not localize properly is based on a figure that shows two cells. In light of the importance of this result, bigger fields with more cells should be shown in the main figure or the whole field should be included as an additional figure attached to a primary figure.

2) The results of the Western blot analyses (Figure 6) should be presented next to the chart or at the end as supplemental data.

3) The following sentences, in the Results section, need to be edited:

“Detailed analysis of the fluorescence signal distribution in all these metabolic conditions revealed that polar localization is systematically associated with the formation of fluorescent clusters.” This sentence is odd; doesn't “polar localization” invoke the existence of “fluorescent clusters”?

“Interestingly, polar localization requires both the absence of oxygen and the establishment of a proton concentration gradient to which the nitrate reductase complex may contribute fuelling the localization mechanism.” A comment about the effect of oxygen on NarG localization should be added here or in the Discussion.

“Altogether, these results are supportive of gene expression being essential for polar localization of the nitrate reductase complex.” This should be rephrased. How about saying that induction of at least one gene during the shift oxic to anoxic conditions promotes polarization of NarG.

“As seen in Figure 7, within the first 10 minutes following nitrate addition, the rate of nitrite production is significantly higher in PR-expressing cells than in control cells, indicating a direct correlation between subcellular localization and electron flux through the nitrate reductase complex. After one hour, the level of nitrite produced is nearly two orders of magnitude higher in PR-expressing cells. We conclude that polar clustering of the nitrate reductase complex results in a higher efficiency of the associated respiratory chains.” This last conclusion is not fully justified. The sentence should state that polar clustering of the complex and/or induction of a ∆pH (or possibly another activity) by the PR promotes respiratory chain efficiency.

---

## [Author Response]

*1) Additional experiments are needed to support the conclusion that “these dynamics are critical for controlling the activity of nitrate reductase*.*”*

*A mutant that does not localize properly and consequently is inactive should be identified*.

We have now included additional data concerning a catalytically inactive but stable variant of the nitrate reductase complex allowing to evaluate whether functionality is essential for polar localization (Results and Figure 5). Our results clearly show that activity of the complex is essential for polar positioning unless a ∆pH is established.

Other mutants may also be considered such as those affecting gene(s) which would be involved in the localization mechanism. However, their search and identification is clearly outside the scope of our manuscript and will require the use of genetic screening for instance. While this is obviously one of the future directions of our work, we have decided not to consider such mutants in our revised version. Furthermore, we consider that their identification is not essential to support our conclusion.

*Blue-Native Gel Electrophoresis should be carried out to test the inference that super-assemblies of respiratory complexes (such as observed for mitochondria) are formed. The co-association of other components should also be tested in these assays as well as by fluorescence and immune-labeling microscopy*.

The main conclusion of our work is that the nitrate reductase respiratory complex is submitted to a spatiotemporal regulation in response to environmental conditions which, in turns, potentiates the electron flux through the associated respiratory chain. Evaluating whether the nitrate reductase complex is engaged in a supramolecular organization as requested is again outside the scope of our work. Furthermore, this would constitute an entire work per se which would hardly facilitate reading of our submitted manuscript. Other alternatives to a supramolecular organization of the respiratory chain can also be considered for functional advantages such as segregation into microdomains as mentioned in the discussion section of our manuscript. Consequently, we have decided not to consider these experiments in our revised version.

*The claim that the changes in localization are quicker than would be expected if changes in gene expression were responsible [even though bacterial genes are induced within minutes of anaerobiosis (see Li et al. 1987 J. Bacteriol. 169:4614-4620)], should be tested by localization upshift experiments in cells pre-treated with a protein synthesis inhibitor. The authors should also examine the localization of proteorhodopsin and confirm the localization of native or functional epitope tagged NarG by immunolocalization in these experiments. Finally, there needs to be a negative control to show that proteorhodopsin does not direct any GFP-tagged membrane protein into bi-polar clusters*.

At first, the text has now been clarified as our concern was to question whether redistribution of the complex in the membrane occurs in a shorter timescale than changes in nitrate reductase content as the result of transcriptional regulation of the *nar* gene expression. There is no doubt that transcriptional up-regulation of *nar* gene expression occurs in a very rapid fashion (see for instance, Partridge et al. JBC 2006 and 2007), but how this is translated into changes in protein content considering the intricate structure of the complex of 500kDa with 8 metal centers was unclear. Additional data are now included in the revised manuscript as requested (see Results and Figure 6). During anoxic-oxic transition, the redistribution of the complex is unaffected by chloramphenicol and occurs within several tens of minutes without significant changes in nitrate reductase content (as evaluated by western-blotting). These data support the conclusion that changes in subcellular localization can occur without changes in complex content. In contrast, during the oxic-anoxic transition, FNR-mediated up regulation of the *nar* gene expression precludes any conclusion. Interestingly, these new data indicate that protein synthesis is required for polar localization under these oxic-anoxic conditions.

Finally, we provided evidence for uniform distribution of the proteorhodopsin (PR) in the membrane using its red fluorescence emission (see subsection headed “Fluorescence microscopy”). The membrane localization of nitrate reductase was already ascertained in Figure 1 by immunoblotting and by its ability to sustain growth under nitrate respiration where its functionality is mandatory. Finally, one of the reviewer concerns was to ascertain that PR expression does not direct any membrane protein into bi-polar clusters. We provide here direct evidence that this is not the case in Figure 6. Indeed, PR expression during the oxic-anoxic transition in presence of Cm was not sufficient to promote polar localization.

*2) Appropriate statistical analysis of data needs to be included for all experiments*.

We totally agree and in the revised version, statistical analyses of data have now been included for all experiments. Specifically, fluorescence image analyses were performed as follow. At least three independent experiments were performed and treated separately, calculating frequency of cells with polar clusters and displaying the data as mean frequency with error bars. As such, a total of nearly 1000 cells have been analyzed for each condition. When appropriate, the non-parametric Wilcoxon/Mann-Whitney statistical analysis was included to compare data. The same procedure was used for western-blotting analyses. The data are represented as mean with error bars of the signal ratio between GFP-labelled nitrate reductase and IscS used as an internal control. Again, when appropriate, Wilcoxon/Mann-Whitney statistical analyses are included to compare data.

*3) The Discussion needs to be expanded and address the question “How do the findings fit into the wider field of bacterial cell biology, metabolic channeling in bacteria and organelles?” (Werner et al. PNAS 2009) rather than speculate on host-pathogen interactions in the gut*.

The Discussion section has now been completely rewritten in the context of cell biology.

[Editors' note: further revisions were requested prior to acceptance, as described below.]

*1) The new result with the inactive mutant that does not localize properly is based on a figure that shows two cells. In light of the importance of this result, bigger fields with more cells should be shown in the main figure or the whole field should be included as an additional figure attached to a primary figure*.

We have now included a Figure 5—figure supplement 1 showing a bigger field with more cells under both nitrate and nitrate/fumarate conditions. We consider that both the quantitative analysis of the results shown in Figure 5 and the fluorescence images (Figure 5 and Figure 5—figure supplement 1) fully address the raised concern.

*2) The results of the Western blot analyses (*Figure 6*) should be presented next to the chart or at the end as supplemental data*.

As suggested, we have now transferred the western-blot analyses to Figure 6—figure supplement 1.

*3) The following sentences, in the Results section, need to be edited*:

*“Detailed analysis of the fluorescence signal distribution in all these metabolic conditions revealed that polar localization is systematically associated with the formation of fluorescent clusters*.*” This sentence is odd; doesn't “polar localization” invoke the existence of “fluorescent clusters”?*

We have now clarified this point and modified the text in several instances. In particular, we have rephrased the above-mentioned sentence as follow: “Detailed analysis of the fluorescence signal distribution in all these metabolic conditions revealed that formation of fluorescent clusters is systematically associated with polar localization”.

*“Interestingly, polar localization requires both the absence of oxygen and the establishment of a proton concentration gradient to which the nitrate reductase complex may contribute fuelling the localization mechanism.” A comment about the effect of oxygen on NarG localization should be added here or in the Discussion*.

*“Altogether, these results are supportive of gene expression being essential for polar localization of the nitrate reductase complex.” This should be rephrased. How about saying that induction of at least one gene during the shift oxic to anoxic conditions promotes polarization of NarG*.

Both concerns are related to the effect of oxygen on nitrate reductase localization. At this stage, the most simple hypothesis supported by our data is to consider that expression of at least one gene is required under anaerobiosis to promote polar localization. As such, we have modified the text to address these points as follow: “Altogether, these results support the idea that induction of the expression of at least one gene during the oxic-anoxic transition promotes polar localization of the nitrate reductase complex”.

*“As seen in*
Figure 7*, within the first 10 minutes following nitrate addition, the rate of nitrite production is significantly higher in PR-expressing cells than in control cells, indicating a direct correlation between subcellular localization and electron flux through the nitrate reductase complex. After one hour, the level of nitrite produced is nearly two orders of magnitude higher in PR-expressing cells. We conclude that polar clustering of the nitrate reductase complex results in a higher efficiency of the associated respiratory chains.” This last conclusion is not fully justified. The sentence should state that polar clustering of the complex and/or induction of a ∆pH (or possibly another activity) by the PR promotes respiratory chain efficiency*.

As recommended, we have modified the above-mentioned sentence as follows: “We conclude that environmental conditions (anaerobiosis and a ∆pH) promoting polar clustering of the nitrate reductase complex result in a higher efficiency of the associated respiratory chains”.